# Spatial structure affects phage efficacy in infecting dual-strain biofilms of *Pseudomonas aeruginosa*

Samuele Testa[1,4], Sarah Berger[1,4], Philippe Piccardi[1], Frank Oechslin[1,2], Grégory Resch[1] & Sara Mitri [1,3]*

Bacterial viruses, or phage, are key members of natural microbial communities. Yet much research on bacterial-phage interactions has been conducted in liquid cultures involving single bacterial strains. Here we explored how bacterial diversity affects the success of lytic phage in structured communities. We infected a sensitive *Pseudomonas aeruginosa* strain PAO1 with a lytic phage Pseudomonas 352 in the presence versus absence of an insensitive *P. aeruginosa* strain PA14, in liquid culture versus colonies on agar. We found that both in liquid and in colonies, inter-strain competition reduced resistance evolution in the susceptible strain and decreased phage population size. However, while all sensitive bacteria died in liquid, bacteria in colonies could remain sensitive yet escape phage infection, due mainly to reduced growth in colony centers. In sum, spatial structure can protect bacteria against phage infection, while the presence of competing strains reduces the evolution of resistance to phage.

---

[1] Department of Fundamental Microbiology, University of Lausanne, CH-1015 Lausanne, Switzerland. [2] Department of Biochemistry, Microbiology and Bioinformatics, Université Laval, Québec City, QC, Canada. [3] Swiss Institute for Bioinformatics, Lausanne, Switzerland. [4] These authors have contributed equally: Samuele Testa, Sarah Berger. *email: Sara.Mitri@unil.ch

L ytic bacteriophage, or simply "phage", are viruses that infect bacterial cells, replicate within them and then lyse them to spread and infect new hosts. Lytic phage are major bacterial predators that are highly abundant in number and distribution, thereby playing a key role in regulating bacterial population dynamics[1]. Despite this potential importance, phage are rarely considered in studies of natural bacterial communities, such as the human microbiome project, or the Earth microbiome project although this is beginning to change[2–5].

Their ability to reduce bacterial populations has also been harnessed as a therapeutic method, in "phage therapy", whereby specific phage targeting a given bacterial pathogen is administered to patients to eliminate infections[6,7]. As we struggle to find solutions to tackle the emergence of antibiotic resistance[8], phage therapy has experienced renewed interest as a possible replacement or complementary treatment to antibiotics.

Although our appreciation of the importance of phage biology is on the rise, the experimental systems used to study phage still limit our understanding of their ecology and evolution in natural environments[9]. Phages are typically studied in liquid cultures in the laboratory using a single phage and a single bacterial strain at a time. On the other extreme of the spectrum, clinical studies have been performed where phage cocktails are administered to animal or human hosts[10–13]. Given all the complexity that such environments bring, it is difficult to explain differences between the results of laboratory and clinical studies[10,11,14,15]. Knowledge at an intermediate scale of complexity is clearly missing. Here, we expand on typical laboratory methods to study two dimensions of environmental complexity that likely matter in real microbial ecology: the presence of other bacterial strains, and life in a spatially structured environment.

Natural communities such as the human microbiome, or soil communities are hugely diverse[16,17], including a large repertoire of phages[3,18–20]. Each of these phages tends to be quite host-specific, killing only a narrow range of bacterial strains (but see[21]). When phage attack a given target strain, we can expect little collateral damage to surrounding strains, and may therefore be tempted to also expect infection of the target to be independent of community structure. However, the presence of insensitive strains has been found to alter treatment outcomes by affecting target strain survival[22–24]. Indeed, Harcombe & Bull[22] have shown that competition with a co-inhabiting species could reduce the ability of the targeted sensitive strain to survive phage attack. Their study considered liquid cultures, however. Since then, it has been shown that the spatial organization of different bacterial strains and species within biofilms can drive social interactions and the evolutionary trajectories of bacterial communities[25,26]. Biofilm-associated bacteria also have a higher survival rate compared to planktonic bacteria[27], particularly when exposed to antibiotics and importantly, also to phage[28]. More generally, phage population dynamics differ radically between liquid bacterial cultures and bacteria growing on solid surfaces[29].

Here we show that both of these factors—the presence of other strains, and spatial structure—separately and combined affect the outcome of phage predation on the pathogen *Pseudomonas aeruginosa*, and its ensuing population dynamics. In particular, we targeted *P. aeruginosa* strain PAO1 with Pseudomonas phage 352 to which it is sensitive, in the presence and absence of a second strain, *P. aeruginosa* PA14 that is insensitive to the phage. Since phage are so specific, we believe the choice of a phage-insensitive strain of the same species[30] to be a realistic one. We assessed the outcome for PAO1 in a well-mixed liquid environment compared to a structured biofilm (colony) growing on a solid agar surface.

We found that in liquid, competition between the two strains could reduce the growth rate of the target strain PAO1, and allow

the phage to infect and eliminate PAO1 without the emergence of resistance. Indeed, evolving resistance to the phage was the only way for PAO1 to survive phage attack in liquid. In contrast, in a biofilm treated with phage, PAO1 survived in the presence of the phage-insensitive strain PA14 without becoming resistant itself. Survival in the face of a phage attack, however, did not depend on PA14 but occurred in all biofilms, regardless of the presence of other strains. Instead, in our setup, slower growth in the colony center appears to be the main mechanism that reduced the ability of the phage to replicate and spread through biofilms containing sensitive bacteria. The presence of PA14 in the same biofilm as PAO1 still had an important effect, however: PA14 made it much less likely for PAO1 to become phage-resistant, and reduced the total number of phage.

## Results

**Inter-strain competition reduced PAO1 survival in liquid**. We first sought to understand how treating a target strain *P. aeruginosa* PAO1 (henceforth PAO1) with Pseudomonas phage 352 in well-mixed liquid cultures is affected by the presence of a phage-insensitive strain *P. aeruginosa* PA14 (henceforth PA14). These liquid experiments involved growing bacteria in 96-well plates containing TSB and inoculated with mixtures of bacteria and phage over a period of 48 h.

In control treatments with PAO1 growing alone, phage treatment resulted in a drop in PAO1 population size after 6 h, after which the population recovered somewhat but not entirely (Fig. 1a). Assays testing for phage resistance (see Methods) revealed that after 24 h of culture, 62 out of 63 tested colonies (98.41%) were resistant to the phage, while after 48 h, 24 out of 24 (100%) were resistant (further statistics on resistance rates are in Fig. 1e, f). As a control, insensitive PA14 cells growing alone were not significantly affected by the phage (Fig. 1b, $P = 0.4$).

Next, we co-cultured the two strains in the absence of phage and found that PAO1 grew worse than when it was alone (~2-log difference in co- versus mono-culture at 24 h), presumably due to competition with PA14 (Fig. 1c). Adding the phage to this co-culture eliminated all PAO1 within 6 h (Fig. 1d). Compared to growing alone then, PAO1 resistance could not emerge when growing with a competitor.

We hypothesized that the presence of PA14 prevented PAO1 from increasing its population size, thereby decreasing its mutation supply and its potential to evolve resistance to the phage and survive the treatment. To test for the effect of population size on resistance evolution, we conducted two experiments. First, we grew PAO1 in the presence of phage with different starting population sizes, while maintaining the multiplicity of infection (MOI) constant at 1 (1 phage for each bacterium). In agreement with our hypothesis, resistance to the phage emerged when the initial population size was greater than $10^4$ CFU/ml (Fig. 1e). Second, we kept the initial population size of PAO1 constant at $10^6$ CFU/ml and varied the starting population size of its competitor PA14 in the presence of phage (MOI = 1). Again, as predicted, phage resistance could emerge when there were fewer competitors, but once the number of competitors at the start exceeded $10^6$ CFU/ml, PAO1 cells were all killed by the phage at the end of 21 hours of co-culture (Fig. 1f). In all cases, PAO1 survival depended on becoming resistant to the phage.

In sum, in liquid culture, competition with a resident strain could prevent our targeted strain from surviving phage treatment, which is consistent with previous research[22].

**Phage infected sensitive PAO1 in mono-culture colonies**. To simulate a setup where a biofilm forms on a solid surface and is later exposed to phage, we first grew the bacteria on a membrane

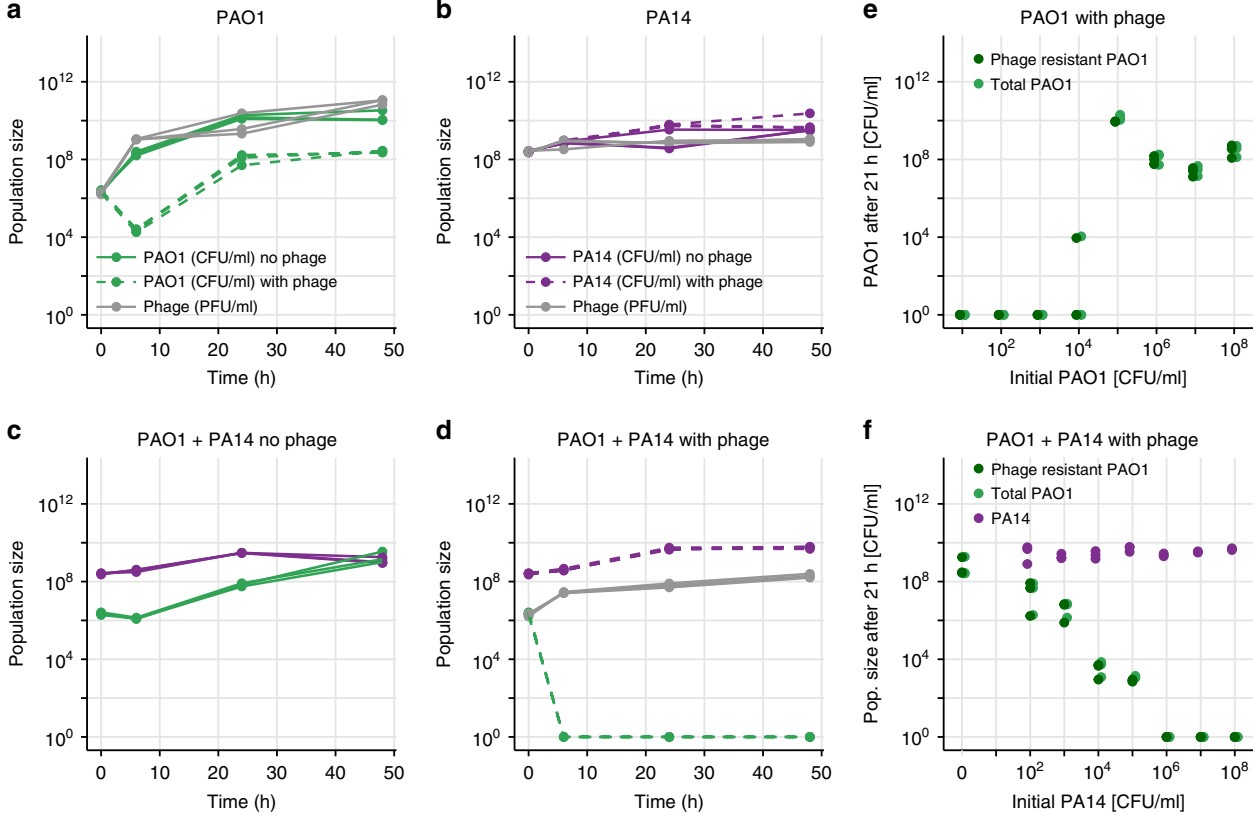

**Fig. 1** Phage efficacy in liquid. **a** Growth of PAO1 (in CFU/ml) in liquid over 48 h. PAO1 grew without phage (solid green lines), but in its presence (dashed green lines) PAO1 decreased then rebounded, resulting in a resistant population (statistics in main text). The phage population (in PFU/ml, gray lines) increased accordingly. **b** PA14 (phage-insensitive), grew similarly in the presence or absence of phage (dashed or solid purple lines, respectively), while phage (gray) remained approximately constant. **c** When PAO1 (green) and PA14 (purple) were grown in co-culture in the absence of phage, PAO1 grew worse than alone. **d** When phage were added to the co-culture, PAO1 population size dropped below the detection limit at 6 h and did not recover. **e** PAO1 was grown together with phage in triplicate at different initial population sizes (MOI = 1). At the end of the experiment, bacteria were plated onto agar plates saturated with phage or not to count the resistant and total population (see Methods). A starting population size greater than ~$10^4$ allowed resistance to emerge. **f** Initial population size of PAO1 was always ~$10^6$, while initial PA14 numbers varied as on the x-axis. Once PA14 became too numerous (greater than ~$10^6$), PAO1 could no longer maintain its population size high enough for resistance to the phage to emerge. Purple, light green and dark green points show population size of PA14, total PAO1 and resistant PAO1, respectively, at 21 h. All panels show raw data coming from three technical replicates

filter placed on LB agar for ~12 h until they had formed a small colony. We then transferred the filter with the 12-h colony onto a new LB agar plate containing an air-dried drop (approximately the diameter of the filter) of either ~$10^6$ phage, ~$10^9$ phage, or no drop as a control. All colonies were left to grow in the presence or absence of the phage for an additional 36 h, approximately (see Methods, Fig. 2a).

In PAO1 mono-culture colonies treated with phage, populations ceased to grow following phage arrival (comparison of CFUs at 12 and 48 h from five experimental repeats with three replicates each, $df = 29$, $P = 0.58$, Fig. 2c, Supplementary Figs. 1, 2, 5, 14), and there were significantly fewer bacteria in the phage treatment compared to the control ($7.96 \pm 6.02 \times 10^7$ with phage, versus $7.51 \pm 3.82 \times 10^8$ without, $df = 23$, $P < 0.001$). Fluorescence microscopy images taken immediately prior to infection and 36 h later showed that colonies treated with phage were smaller in diameter than non-treated colonies, with the fluorescent cells still visible in the center of the colony (Fig. 2b, middle column, Supplementary Fig. 3). In the colonies that had been treated with phage, resistance to the phage was detected in 14 out of 15 colonies across five similar experiments, with resistant cells forming between ~0.04 and 20% of the total population at low (~$10^6$ PFU/ml) initial phage dose (Fig. 2d, Supplementary Fig. 4). At high initial phage dose (~$10^9$ PFU/ml),

the majority of surviving cells were found to be resistant to the phage, but a sub-population of sensitive cells survived in all replicates (Supplementary Fig. 2f).

**Infection and phage resistance were mainly at colony edges**. We wondered why so many sensitive cells survived and where in the colony resistance had occurred. To answer this question, before harvesting the colonies for quantification, we touched an inoculation loop in the center of the colony, resuspended its contents in phosphate-buffered saline (PBS) and plated the suspension to quantify the number of resistant and sensitive cells, as well as phage (see Methods). We found no resistant cells in the center of any of the colonies (Fig. 2d, Supplementary Fig. 4). Phage were nevertheless detected in the center, but the ratio of phage to uninfected cells was significantly lower than in the colony as a whole (PFU/CFU of $0.99 \pm 0.27$ in the center and $276.5 \pm 11$ in the whole colony, paired $t$-test, $P < 0.001$, Fig. 2e). This suggests that resistance arose closer to colony edges where most cellular growth occurred[31], and where phage titers were highest. The phage could still spread to the center of the colony, but only infected a proportion of cells. Further evidence that some cells in the center were infected was that after washing to remove phage and plating on fresh agar, most cells lysed (Supplementary Fig. 6), resembling

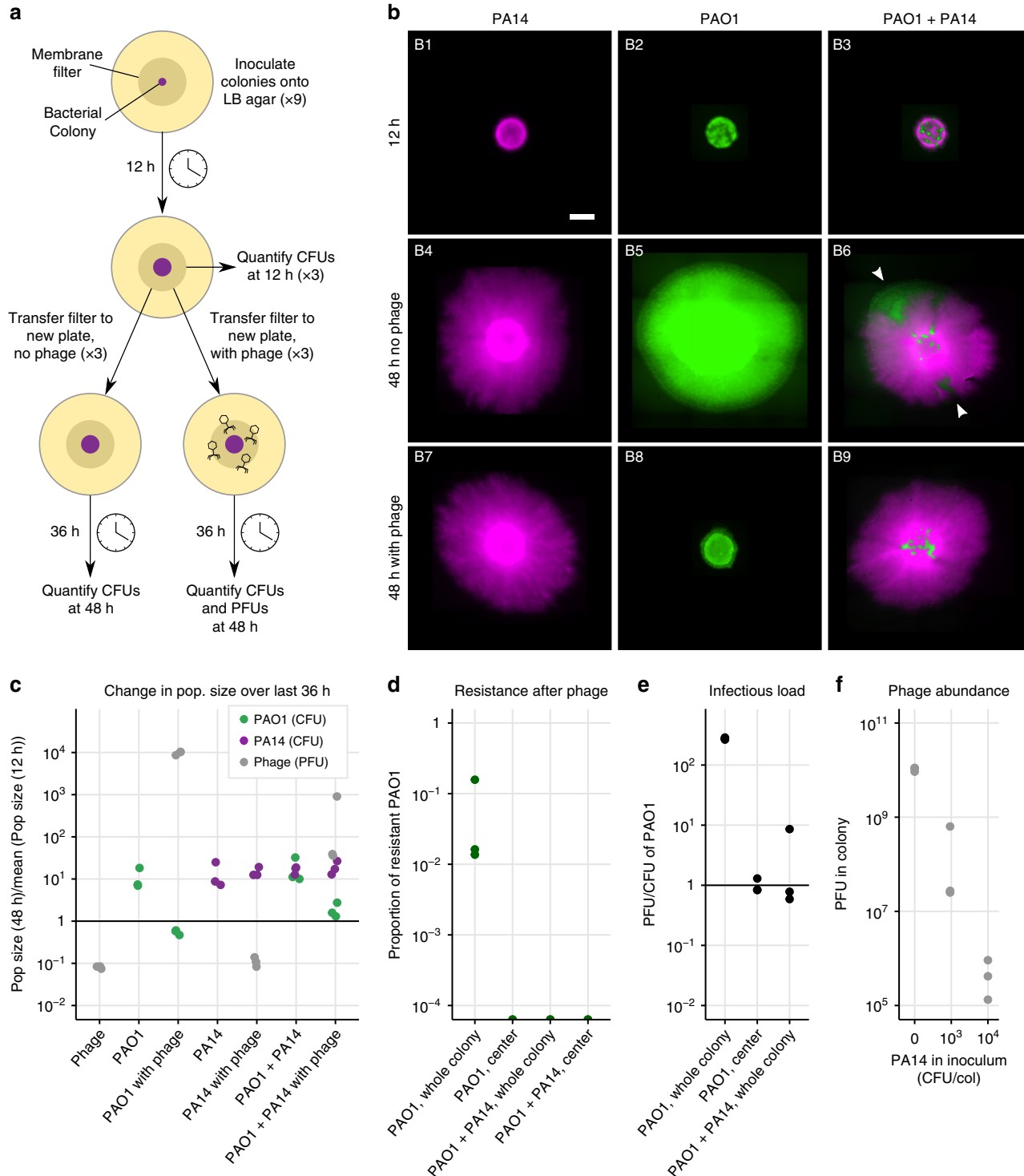

**Fig.** (implied by panels a–f)

"pseudolysogeny" or "hibernation", which occurs in starved cells in stationary phase or persister cells, where phage DNA accumulates in the cell[29,32–36]. However, transmission electron micrographs of the fixed colony revealed many intact cells containing phage with assembled capsids that had not yet lysed, in addition to some debris from lysed cells (Fig. 3). While phage were able to assemble—contrary to expectations for pseudolysogeny—the presence of unlysed and uninfected cells suggests a delay in lysis, which may explain why phage could not spread further and increase their numbers in the colony center.

**Phage penetration into PA14 colonies was limited**. In contrast, PA14 (the phage-insensitive strain) mono-culture colonies were indistinguishable with and without phage treatment (Fig. 2b, c, *t*-test CFUs with and without phage, df = 2.6, P = 0.87, Supplementary Figs. 1, 5). On sampling the colony centers, we never found phage in any of the colonies treated with a low phage dosage, but detected a few at the high initial dose of phage (on average 1 phage to every 863 PA14 cells). This suggests that physical diffusion of the phage from the agar into PA14 colonies is very limited. Indeed, total phage populations fell to 11 ± 2.8% of

**Fig. 2 Phage efficacy in colonies. a** All colonies (PAO1, PA14 or PAO1 + PA14) were first grown by inoculating cells onto a membrane filter placed on 0.1x LB agar in 9 replicates. After ~12 h, three filters were destructively sampled to quantify CFUs, three were transferred to fresh 0.1x LB agar containing a dried 50 µl drop of phage containing ~$10^6$ PFU/ml, and three to fresh 0.1x LB agar containing no phage. After ~36 h the remaining 6 filters were harvested to quantify CFUs and/or PFUs (see Methods). **b** Fluorescence microscopy images of colonies at 12 and 48 h. PA14 are tagged with mCherry (purple) and PAO1 with GFP (green). Sectors that formed in untreated mixed colonies (white arrowheads, B6) were absent in the phage treatment (B9) suggesting that phage kill cells at the actively growing colony edges, while cells in the center survive (Supplementary Fig. 3 shows a similar experiment). The scalebar in B1 represents 2 mm. **c–f** Data coming from triplicate colonies using unlabelled PA14. These data do not correspond to the images in **b**, whose quantification was less precise (see Supplementary Fig. 5) because PA14-mCherry were difficult to distinguish from PAO1-GFP (identical drug resistances). **c** The ratio of population sizes at 48 and 12 h (see Supplementary Fig. 1 for growth curves). The mean at 12 h was used due to destructive sampling. Phage decrease PAO1 (green), insensitive PA14 grow similarly across conditions (purple), and phage decrease in the absence of PAO1 and increase in its presence.
**d** Phage-resistant PAO1 were found in whole colonies of PAO1, but not in the centers of PAO1 colonies, nor in colonies mixed with PA14. **e** The ratio of PFUs to CFUs was lower in the colony centers, and in whole mixed colonies compared to whole colonies of PAO1 alone. **f** Phage abundance at 48h is inversely proportional to the initial abundance of PA14 in the colony inoculum. PAO1 was constant in all inocula at $10^2$ CFU/col. Note that $10^6$ PFU were added after 12 hours of growth and the plot shows the final PFU values

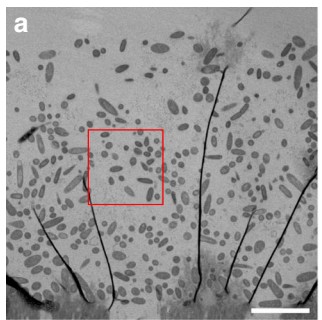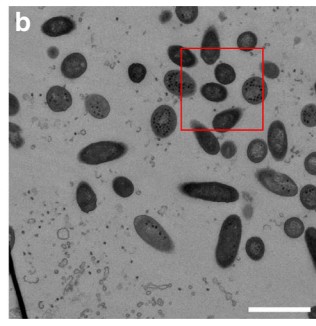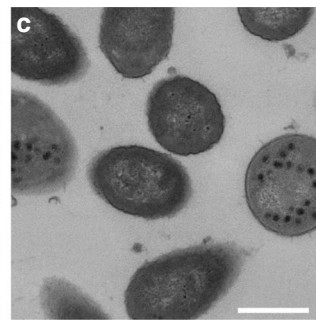

**Fig. 3 Transmission electron micrograph of an infected PAO1 colony. a** A slice through a colony, with the filter visible at the bottom, and an empty zone at the top. We cannot be certain of the location of the slice within the colony. The dark black lines are an artefact due to sample preparation. The red box shows the location of panel **b**. The scalebar indicates 5 µm. **b** Phage are visible inside and around the cells. Note that cell debris from lysed cells is also visible in the bottom left corner. The red box shows the location of panel **c**. The scalebar indicates 1 µm. **c** Example of some cells containing assembled phage, and others that were free of phage. The scalebar indicates 400 nm. Cells containing phage had not lysed, suggesting delayed lysis, but not "pseudolysogeny", where we would not expect assembled phage capsids

their original size in PA14 colonies over the 36 h, which we suspect is due to toxicity of LB to phage[37] or temperature sensitivity, given that phage populations also fell to 8.1 ± 5.4% in the absence of any bacteria (Fig. 2c, *t*-test with and without PA14: $P = 0.21$). To determine whether phage could attach to PA14 cells, we performed an adsorption assay in liquid, and found that after 5 min of exposure to bacteria, phage only attached to PAO1 cells, but not PA14 (Supplementary Fig. 7).

Taken together, in single-strain colonies we observed that PAO1 death and the emergence of phage resistance occurred mainly at the edges of the colony where cells were more actively growing. Only very few phage could spread into insensitive PA14 colonies at high phage titer, suggesting limited physical diffusion into PA14 colonies. Instead, cycles of attachment, infection and lysis allowed phage to propagate deeper into colonies of sensitive PAO1. Phage could therefore infect some, but not lyse all PAO1 cells at the colony centers, where they were less metabolically active.

**Phage infected sensitive PAO1 in mixed colonies**. Knowing that phage could not diffuse much into PA14 colonies, we next asked how the presence of this insensitive strain would impact the survival of the targeted PAO1 within a colony containing both strains and treated with phage. We repeated the experiment (Fig. 2a) with a mixture of both PAO1 and PA14 at an initial ratio of 1:10, such that an approximate 1:1 ratio was reached on phage exposure after 12 h (Supplementary Fig. 1).

As in the phage-treated PAO1 mono-culture colony, PAO1 in the treated mixed colonies did not increase significantly following phage treatment ($df = 17$, $P = 0.47$, Fig. 2c, Supplementary Fig. 2), and the phage treatment significantly reduced PAO1 cells compared to the untreated control ($df = 17$, $P < 0.001$), demonstrating significant bacterial infection by phage. In addition, microscopy showed that patches of PAO1 (white arrowheads in Fig. 2b, B6) were absent from the edges of the colonies treated with phage (Fig. 2b, B9). These data suggest that as in PAO1 colonies, cell lysis occurred at the actively growing edges.

We observed two important differences between mono- and co-culture colonies, however. First, as in the liquid co-culture, phage resistance was much less likely to emerge, with no phage resistance in mixed colonies infected with the low phage dose (Fig. 2d, Supplementary Fig. 4), while at high infective dose, 0.6 ± 0.3% of cells were resistant compared to the vast majority in the mono-culture colonies (Supplementary Fig. 2). Second, co-culture colonies contained a lower infectious load (fewer phage per sensitive bacteria) compared to mono-culture colonies at the end of the experiment ($df = 20$, $P < 0.05$, Fig. 2e, Supplementary Fig. 2), indicating that phage could replicate less in the presence of PA14. To further verify this, we increased the number of PA14 in the colony inocula while keeping PAO1 constant, and found that phage abundance in the colony follows a strong negative correlation with initial PA14 abundance (Spearman's $\rho = 0.91$, $P < 10^{-7}$, Fig. 2f). Both these findings can be explained by what we observed in the images: since only a small proportion of the edge of a mixed colony was made up of PAO1 cells (Fig. 2b, white

arrows), the effective population size of PAO1 (i.e., number of growing cells) was smaller in the presence of PA14 than in its absence[31,38], making the emergence of resistance less likely (Fig. 1d), and keeping phage populations that infected them smaller (Fig. 2f).

Given that the phage mainly infected PAO1 in the edges of both mixed and mono-culture colonies, we hypothesized that we would find phage refuges containing uninfected cells in both conditions, regardless of the presence of PA14.

**All colony centers contained refuges of uninfected PAO1.** To search for uninfected PAO1 in different areas of the colonies, we sampled the mono-culture and mixed colonies previously exposed to ~$10^6$ PFU/ml by touching them with sterile toothpicks in four locations (Fig. 4a), resuspending the cells and phage on the toothpicks in PBS and then spotting a drop of each suspension onto different agar plates to quantify the density of PAO1, phage, and PAO1 cells that had become resistant to phage (see Methods). To analyze these data (Supplementary Figs. 8, 9), we imaged each drop after 15 h of growth and processed the images (Fig. 4b) to quantify the density of healthy PAO1 cells and phage plaques in each position. A standard curve mapping this density to population size is shown in Supplementary Fig. 10.

In agreement with previous experiments, we only observed resistant PAO1 cells in samples coming from the mono-culture colonies, and these were detected in positions sampled further away from the colony center (Fig. 4c, Supplementary Fig. 11, triangles). Moreover, in line with our previous observation that PAO1 at the edge were killed by phage, sampling at the edge of the mixed colonies yielded very few PAO1 cells, and also very few phage (Fig. 4c, Supplementary Fig. 11 black dots close to origin in right panel).

For all remaining samples (where sensitive PAO1 and phage density were >0.1), phage density correlated negatively with the density of PAO1 (Pearson $\rho = -0.9$, $P < 10^{-10}$), as one would expect. If PA14 had a protective effect, we would expect to find fewer phage-protected refuges in the center of mono-culture compared to mixed colonies. Instead, 35% of the samples from the mono-culture colonies and 20% from the mixed colonies had sensitive PAO1 cells close to the center that were completely uninfected by the phage (top left points in Fig. 4c, with PAO1 density > 0.25). This supports the presence of refuges in the centers of all colonies, and rejects the hypothesis of phage-free refuges being caused solely by the presence of PA14.

This assay can be seen to represent a scenario where cells would have a chance to leave a biofilm and reseed a new environment. Cells from the refuges that were uninfected by phage would begin to grow and start new, healthy colonies (see also Supplementary Fig. 6).

**Growth arrest is the main factor preventing phage infection.** To explain why many sensitive cells in the colony centers remained uninfected with or without PA14, we put forward two hypotheses that we tested next: first, that cells in the center of any colony could avoid phage infection because of a lack of growth; and second, that phage-resistant cells (not only insensitive PA14, but also newly emerging resistant cells) could create phage-free refuges in colonies by preventing phage spreading through reduced phage amplification. Accordingly, we repeated the experiment of Fig. 2a with two new conditions: in the first, we used phage-sensitive PAO1 cells (wild-type) but after the 12 h of growth, we moved them onto agar lacking LB and containing Ethylenediaminetetraacetic acid (EDTA) to arrest bacterial growth, forcing them into stationary phase[39]; and in the second we combined our wild-type PAO1 with 10× more of a phage-

resistant PAO1 strain res1, isolated from the experiments described above (see Methods). These resistant mutants were found to be lacking the *galU* gene (Supplementary Fig. 12), resulting in a loss of lipopolysaccharides (LPS) and preventing phage attachment, as observed in previous work[40,41] and verified by an adsorption assay (Supplementary Fig. 7). A large fitness cost was associated with the loss of LPS, as shown in a co-culture of wild-type and mutant PAO1 res1 without phage (Fig. 5a).

The growth-arrested colony grew slightly (by 131 ± 33.4% over 36 h), and the phage increased 19.9-fold, approximately three orders of magnitude less than in a PAO1 colony growing on LB agar (Fig. 5a). Even though the phage replicated, they were not detected in the colony centers (0 in all three replicates) (Fig. 5c). In contrast, phage were found in the centers of colonies of the mixed phage-sensitive and -resistant PAO1, at an infectious load that was similar to the mono-culture colonies (Fig. 5c). In other words, even though the colony started with 10 × more resistant cells, phage could still easily infect the sensitive bacteria and spread through the colonies (Fig. 5a, c). It is therefore unlikely that a rare mutant arising in a wild-type colony would provide protection to the sensitive cells, at least in part due to their reduced fitness. Moreover, the reduced competitiveness of PAO1 res1 compared to the wild-type PAO1 meant that phage populations did not even suffer in its presence and PAO1 could become resistant (Fig. 5b). This stands in contrast to the more competitive PA14, with which phage titers correlated negatively and resistance did not evolve (Fig. 2d, f). No phage were detected in the centers of control colonies containing only resistant PAO1, which were comparable to PA14 mono-culture colonies (Fig. 5c).

The data from these treatments support our previous conclusion: to reach the center of *P. aeruginosa* colonies, Pseudomonas phage 352 needed to attach to the surfaces of bacterial cells, and infect them while they were actively growing and dividing. Since phage-free refuges were observed in some mono-culture colonies where no resistance was detected (Supplementary Fig. 9), and since phage could easily infect wild-type PAO1 in the presence of a large initial population of resistant PAO1, we conclude that growth arrest played a more important role in protecting sensitive bacteria against phage infection compared to being surrounded by resistant or insensitive cells that phage could not attach to or infect. However, only a strong enough competitor (PA14, but not PAO1 res1) could reduce resistance emergence and phage population size.

**Discussion**
In spatially organized biofilms, resistant bacteria can protect sensitive ones against different forms of environmental assault, such as antibiotics[42–45] or predators[46]. We were curious whether such cross-protection would also be observed against phage in spatially structured biofilms. We found little evidence for this: an insensitive strain (or a newly emerging resistant strain) provided no additional protection to sensitive bacteria from the phage (Fig. 6c). Instead, sensitive bacteria avoided phage infection as their growth rate decreased within the biofilm where nutrients were less accessible. We nevertheless found an important effect of the presence of the insensitive strain: Sensitive bacteria were less likely to develop resistance to the phage because inter-strain competition prevented them from growing to a sufficiently large effective population size (Fig. 6b). Competitors thus reduced the likelihood of resistance evolution as well as phage population size.

To our knowledge, two studies have investigated cross-protection by infecting co-cultures of resistant and susceptible bacteria with phage. In the first, Tzipilevich et al.[23] found that rather than cross-protection, a sensitive strain of *Bacillus subtilis* actually conferred temporary phage-sensitivity to its previously

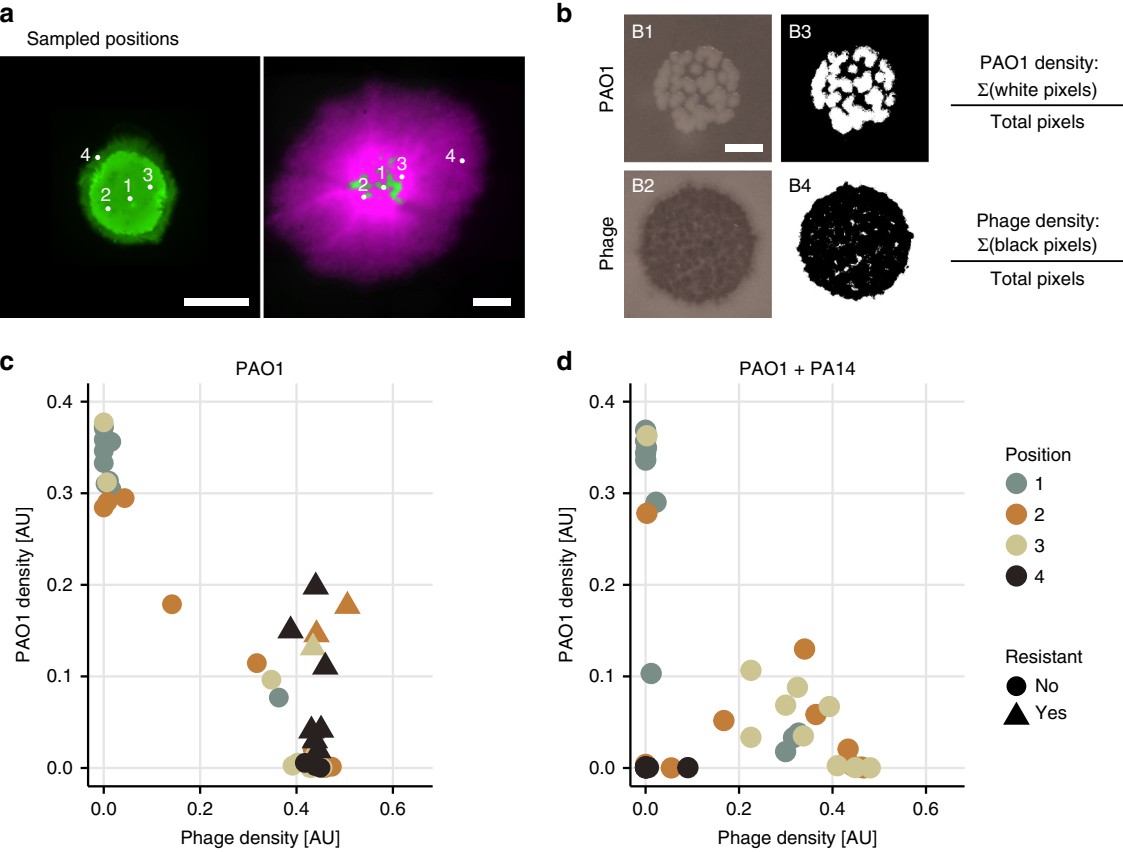

**Fig. 4** Sampling colonies to determine co-occurrence patterns of phage and bacteria. **a** The white dots indicate where we sampled in each colony using sterile toothpicks (same images as in Fig. 2b). Position 2 and 3 were approximately equidistant from the center (position 1). The scalebars indicate 2 mm. **b** The toothpick-attached cells and phage were resuspended and a small drop of the suspension placed on LB agar containing gentamicin, and on soft LB agar containing PAO1. B1 and B2 are two representative images of these drops after ~15 h at 37 °C. The scalebar indicates 3 mm. To quantify the density of bacteria and phage, we applied a threshold to the images (B3 and B4) and then calculated the proportion of white and black pixels in each picture respectively. These values are plotted in panels **c** and **d**. **c**, **d** Density of PAO1 and phage in each sample. Each dot or triangle corresponds to a sample in one position in one colony. **c**, **d** show samples taken from 10 PAO1 and 10 mixed colonies, respectively (4 × 10 = 40 points on each plot). Resistance was determined by similarly thresholding images of drops grown on LB agar with gentamicin and saturated with ~$10^{10}$ phage (see Supplementary Figs. 8, 9 for the full data set). The different colors represent the positions sampled as shown in panel **a**. Sampled positions are approximate

resistant neighbor. This happened through the horizontal transfer of phage attachment molecules from lysed sensitive cells to intact resistant ones. In contrast, Payne et al.[24] demonstrated that cross-protection against phage T7 can occur between two strains of *E. coli*, where one harbored a CRISPR-based resistance. Cross-protection was observed both in liquid and on a bacterial lawn. One key difference to our study is that their CRISPR-immune cells removed the phage from the environment through adsorption and degradation, and then stopped growing, whereas in our system, phage did not even attach to insensitive PA14 or resistant PAO1 cells. These cells simply did not seem to interact with the phage.

A number of mechanisms have been proposed to explain our observation that sensitive bacteria are more likely to survive phage attack in biofilm than in liquid[28,47]: bacteria may reduce the expression of their phage receptors[48]; a high bacterial density or large molecules, such as exopolysaccharides can reduce phage diffusion[49,50]; finally, if bacteria slow down growth as nutrients are depleted, phage replication also slows down[28,35,36,49,51–56]. Our data support this latter model whereby growth arrest in colony centers greatly reduces the ability of phage to amplify, lyse cells and spread into the center (Fig. 6b). The uninfected, phage-sensitive cells that remain can then potentially seed new, healthy bacterial colonies, if dispersed.

Overall then, whether bacteria survive a phage attack appears to strongly depend on growth conditions and on properties of the phage and bacteria, including their resistance mechanism. First, phages differ in their ability to infect stationary-phase bacteria[36,57]. Phages of different sizes or hydrophobicities may be better or worse at diffusing through biofilm[58]. Second, we expect growth conditions to affect the outcome. For example, providing bacteria with constantly replenishing nutrients would limit growth arrest. Third, since PAO1 and PA14 compete with one another, they tend to separate in space. Strains that rely on each other to grow have instead been shown to remain mixed in colonies[59,60]. Increased mixing may then increase cross-protection against phage. Taken together, we expect reduced phage infection in bacterial biofilms to occur either when cells are in stationary phase, or when they are surrounded by resistant cells that adsorb or degrade the phage.

Our experimental setup is clearly very simple and is not intended to predict phage therapy outcomes. Some bacterial infections may contain multiple strains, but others in the blood stream, for example, will consist of clonal populations. And while *P. aeruginosa* strains do tend to co-exist in lung infections[61], competition between them means that they may exclude each other. It is also unclear whether these results will differ for phages other than Pseudomonas phage 352. Nevertheless, our results

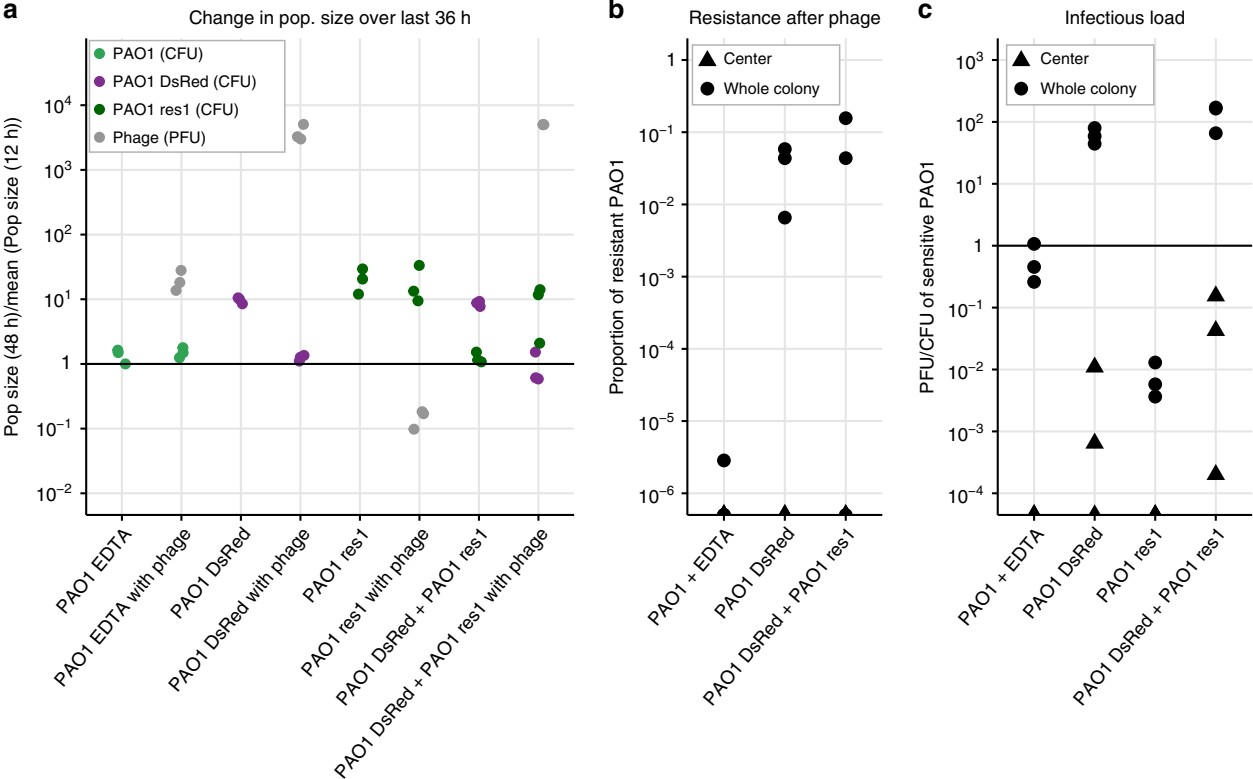

**Fig. 5** Phage infection of PAO1 growing on EDTA, and a resistant PAO1 isolate in mono-culture, and co-cultures of sensitive and resistant PAO1. **a** Ratio of population sizes at 48 and 12 hours (as in Fig. 2, see Supplementary Figs. 13, 14 for growth curves). PAO1 did not grow on EDTA and was hardly infected by phage; PAO1 DsRed behaved as PAO1 GFP (compare to Fig. 2c); resistant PAO1 (res1) grew and was not infected by phage; PAO1 DsRed outgrew PAO1 res1 in co-culture, but could be infected by phage similarly to in mono-culture. **b** Resistance to phage (compare to Fig. 2d) never arose in colony centers. Almost no resistance was observed in the growth-arrested PAO1 on EDTA, while PAO1 DsRed became resistant regardless of the presence of PAO1 res1. **c** The infectious load (compare to Fig. 2e) was relatively low in PAO1 on EDTA and the resistant PAO1 (res1). Infectious load appeared similar whether PAO1 DsRed was growing alone or together with the resistant PAO1. Overall, PAO1 res1 did not protect sensitive cells against phage, but growth arrest (EDTA) did

reveal a context-dependency that may explain why such large discrepancies are observed between laboratory results and in vivo trials of phage therapy[10,11,14,15]. We also confirm some commonly observed phenomena. First, biofilms—a typical mode of growth in an infection—appear to be more difficult to treat with phage compared to liquid cultures. Second, we observe a fitness trade-off between phage resistance and fitness (Fig. 5, Supplementary Fig. 14), which is in agreement with phage-resistant pathogens typically being less virulent than wild-type strains (reviewed in ref. [13]). Finally, we show that the presence of a competitor can reduce the population size of a target strain, which reduces its evolutionary potential as well as phage population sizes. Future research should explore these dynamics within a host to understand whether our findings pose a problem for phage therapy.

Finally, we highlight the importance of spatial structure for the ecology and evolution of microbial populations. In a liquid environment, phage may drive sensitive strains locally extinct, potentially destabilizing the bacterial community. In a multi-strain biofilm, phage may instead generate diversity through uneven infection, which creates local areas of either phage-resistant, phage-infected or phage-protected bacteria (Fig. 4c, Fig. 6), each subject to different selection pressures[51]. In turn, phage have access to different bacterial niches[51,62]. The resulting co-evolutionary dynamics mean that spatially organized bacteria-phage populations, which are likely to be the norm in many environments, may be key to maintaining the diversity, stability and the evolvability of microbial communities.

## Methods

**Bacterial strains, phage, media, and culture conditions**. Experiments were performed with two different strains of *Pseudomonas aeruginosa*: strain PAO1 modified with a miniTn7 transposon containing a GFP (shown in green) or DsRed marker, which was susceptible to a specific phage, and strains PA14 (PA14-WT) or modified with a Tn7 transposon containing an mCherry marker (PA14-mCherry, shown in purple), which were both insensitive to this same phage. Both transposons contained a gentamicin resistance gene. The two PAO1 strains and PA14-WT were kindly provided by Kevin Foster. The phage used for this study was Pseudomonas phage 352, Myoviridae morphotype A1, previously $\phi 14$[63,64] (received from D. Haas, J.-F. Vieu, E. Ashenov, and R. Lindberg). We chose this phage among 14 that we tested because it caused cell lysis and plaque formation in PAO1-GFP but not the two PA14 strains, which were entirely insensitive.

Overnight cultures were grown in tryptic soy broth (TSB; Bacto$^{TM}$, Detroit, MI, USA) at 37 °C, shaken at 200 rpm. Before each experiment, the optical density ($OD_{600}$) of the overnight cultures of PAO1-GFP, and either PA14-mCherry or PA14-WT strains (depending on the experiment) was measured with a spectrophotometer (Ultrospec 10, Amersham Biosciences). Bacterial overnight cultures were then inoculated into Erlenmeyer flasks (100 ml) containing 20 ml of TSB to obtain a standardized $OD_{600}$ of 0.05. Bacterial cultures were grown in a shaking incubator at 200 rpm and 37 °C for 3 h to obtain bacteria in exponential phase with a final density of approximately $10^8$ CFU/ml at the beginning of each experiment (this was very consistent, as can be seen in data at t = 0, which was measured by plating). These cultures were then diluted in PBS to the desired starting population size.

**Quantifying bacterial and phage populations**. To quantify bacterial colony-forming units (CFU) and plaque-forming units (PFU) of phages, in liquid assays, CFUs and PFUs were measured directly, while for colonies, bacteria and phage (if applicable) were first extracted from the filters and suspended in PBS (see below). Suspensions coming from phage-treated liquid cultures or colonies were centrifuged at 8000 rpm for 15 min at 4 °C. Following centrifugation, the supernatant containing phage was kept in the fridge at 4 °C and later used to measure PFUs.

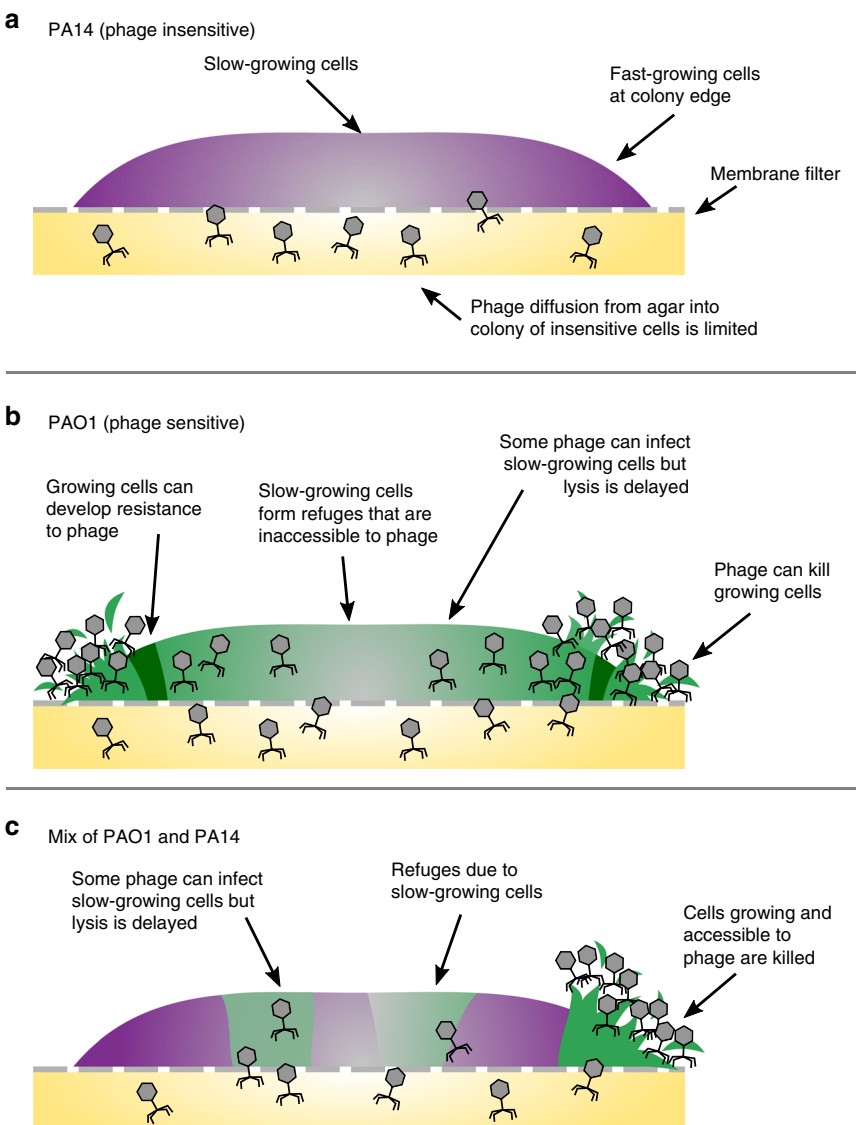

**Fig. 6** We propose a model for how Pseudomonas phage 352 infects colonies of single and mixed PAO1 and PA14 strains. Each drawing shows a cross-section of a bacterial colony, where higher bacterial growth rates are represented by solid colors and slower-growing bacteria by gray. **a** Phage infection and penetration into colonies of insensitive PA14 was limited. The same was found for PAO1 that had acquired resistance. **b** PAO1 colonies were increasingly infected towards the colony edges, correlating with growth rate[31]. Phage resistance (dark green) was observed closer to the edges where growth and infection were occurring. Slow-growing cells toward the colony center formed phage refuges. There, phage infected some cells of which only a subset was lysed. **c** In a mixture of sensitive and insensitive bacteria, insensitive cells reduced phage abundance overall, but phage-free refuges were mainly due to slow growth in the center. The emergence of phage resistance was limited in the presence of PA14

The centrifuged bacterial pellet was resuspended in either 200 µl (liquid) or 1 ml (colonies) of PBS, and then washed three times with 1 ml of fresh PBS at 8000 rpm, 4 °C for 5 min to remove all the potential phages remaining in the pellet.

CFUs were quantified by serially diluting all cell suspensions (from liquid cultures or colonies, with or without phage) from $10^0$ to $10^{-7}$ in PBS, and spreading 10 µl drops in lines across Tryptic Soy Agar (TSA) or Luria-Bertani (LB) agar plates. After 15 h in a 37 °C incubator, colonies were counted at the most appropriate dilution. To distinguish the two *P. aeruginosa* strains, co-cultures were plated onto TSA or LB agar plates to count non-fluorescent PA14-WT CFUs and onto LB agar plates containing 10 µg/ml of gentamicin to count only PAO1-GFP CFUs. In experiments where PAO1-GFP and PA14-mCherry, or PAO1-GFP and PAO1-DsRed were co-cultured, both strains were resistant to gentamicin, and were only plated onto TSA or LB agar and distinguished by their fluorescence. PFUs were quantified similarly, except that drops were spread in lines across 20 ml soft LB agar (30 g/L LB + 7 g/L agar) mixed with 300 µl of PAO1 overnight culture and allowed to dry for 1 h in a laminar flow hood. For the treatments involving phage in colonies, the whole agar was also collected and put in 50 ml falcon tubes containing 10 ml of PBS, well-shaken, centrifuged for 15 min at 4000 rpm at 4 °C and the supernatant containing phages further diluted in PBS to count the PFUs. Phage concentrations from the filter including the colony, the agar and the

touched colony center (see below) were summed up to obtain the final PFU/colony value.

The rate of resistance of PAO1 to the phage was calculated in two ways. In the first method, 10 µl of a solution containing $10^{10}$ PFU/ml was streaked in a straight line across an LB agar plate. Then, having previously plated PAO1 to count CFUs, individual colonies were picked and streaked in parallel lines perpendicular to the line containing phage, and the plate incubated for 15 h at 37 °C. Picked colonies that resulted in solid lines across the length of the plate were classified as resistant, while bacteria in lines that were truncated where the phage had been spread were considered to be sensitive. For the second method, we plated cultures on TSA plates on which we had previously spread 500 µl containing approximately $10^{10}$ PFU/ml of pre-absorbed phages, and allowed to dry. If PAO1-GFP were growing in co-culture with PA14-WT, plates additionally containing 10 µg/ml gentamicin were used. To evaluate resistance rates, the CFUs/colony of PAO1-GFP growing on plates saturated with phages (resistant) was then compared to the CFUs/colony growing on plates with no phage (total uninfected).

**Phage treatment in liquid cultures**. A 96-well plate was filled with 200 µl of TSB in each well, additionally containing $10^6$ CFU/ml PAO1-GFP or $10^8$ CFU/ml

PA14-WT alone, or together with or without $10^6$ PFU/ml of phages (MOI(PAO1) = 1). The 100-fold difference in starting population sizes was based on pilot experiments, where we found that the two strains could co-exist at that initial ratio (Supplementary Fig. 15). In PA14 mono-cultures, $10^8$ PFU/ml were inoculated (MOI(PA14) = 1). Initial population sizes of bacteria and phage were quantified prior to mixing. Each condition (PAO1-GFP alone, PA14-WT alone, and the co-culture) was performed in triplicate. The plate was then put in a Tecan Infinite 200 PRO plate reader at 37 °C under agitation for 48 h. After 6, 24, and 48 h, the samples were transferred into Eppendorf tubes, washed, serially diluted and plated as described above.

**Quantifying phage resistance rates in liquid**. To understand the role of population size on resistance emergence, two experiments were performed (Fig. 1e, f). In the first, a 96-well plate was filled with 10 up to $10^8$ CFU/ml of PAO1-GFP, with 10-fold increases, together with phage to achieve an MOI(PAO1) = 1 in 200 μl of TSB. We grew the bacteria for 21 h at 37 °C under agitation in the plate reader, and then assessed phage resistance rates and total population sizes as described above. For the second experiment, a 96-well plate was filled with $10^6$ CFU/ml of PAO1-GFP and phages at an MOI(PAO1) = 1 in 200 μl of TSB, to which we added increasing amounts of PA14, starting at $10^2$ up to $10^8$ in 10-fold increments. Bacteria were again grown in the plate reader for 21 h at 37 °C under agitation, at the end of which we assessed phage resistance rates and total population sizes of both strains as described above.

**Colony experiments and phage treatment**. To grow bacteria in a colony, liquid cultures were prepared and a drop spotted onto a membrane filter (Isopore® Membrane, 0.2 μm PC membrane, GTTP02500, Merck) previously placed in the centre of agar plates containing 0.1x LB (1 g/L of tryptone (ThermoScientific™ Oxoid™ Tryptone), 0.5 g/L of yeast extract (ThermoScientific™ Oxoid™ Yeast Extract Powder), 10 g/L of NaCl (ACROS Organics™, 99.5%), and 15 g/L of agar (Bacto™ agar solidifying agent, BD Diagnostics)). Liquid cultures of the two strains were prepared as described for the liquid experiments and diluted in PBS to obtain a final concentration of $10^4$ CFU/ml of PAO1-GFP and $10^5$ or $10^6$ CFU/ml of PA14-WT (for a ratio of 1:10 or 1:100, respectively). 100 μl of each strain were mixed together, or with 100 μl of PBS for the mono-culture colonies. A 2 μl drop of the mixture was then spotted onto the filter. Nine replicate plates were prepared for each condition (e.g., PAO1-GFP, PA14-WT, and the mixture of both), and incubated at 37 °C. After 12 h of incubation, three replicates were removed and destructively sampled in order to count the CFUs of both strains by removing the filters from the agar using sterile tweezers and placing them in tubes containing 3 ml of PBS. The tubes were extensively vortexed to remove and resuspend the colonies in the PBS, the filters removed and the bacteria plated to count CFUs as described above. Among the six remaining replicates, three were placed onto new 0.1x LB agar plates without phage and the three others were placed onto new 0.1x LB agar plates pre-absorbed with a 50 μl drop containing ~$10^6$ or ~$10^9$ phages (diameter similar to filter diameter) depending on the experiment, and incubated at 37 °C. After ~36 h, to quantify phage infectious load in the colony center of phage-treated colonies, we touched a sterile, plastic inoculation loop to the top center of each colony (without going deep enough to touch the filter) and resuspended its contents in 1 ml of PBS. We then quantified CFUs and PFUs of this suspension as described above. The Isopore® filters with the remaining majority of the colony were then carefully removed with sterile tweezers, resuspended in 3 ml of PBS, and the suspension used to quantify CFUs and PFUs as described above (the phage-treated colonies with the centrifugation step described previously). Finally, the remaining agar was used to quantify PFUs as described above. For the experiments where we arrested the growth of PAO1 after 12 h, agar plates were prepared containing 10 g/L of NaCl, 15 g/L of Bacto™ agar and 0.05 mM of EDTA. These plates were either spotted with a drop containing ~$10^6$ phage or no drop, and the filters transferred onto them as described above.

**Phage adsorption test**. To test whether phage could adsorb to the different strains, we prepared (on ice) a solution containing ~$10^6$ bacteria (either PAO1, PA14, PA14-mCherry, or PAO1 res1) and added ~$10^6$ phage to each. We quantified the PFUs in the starting inoculum of phage as described above. After 5 and 10 min on ice, we filtered 2 ml of the suspension using 3 ml Omnifix® syringe filters with a pore size of 0.22 μm (Cobetter®), and quantified the PFUs in the supernatant as described above. We worked on ice because at room temperature or 37 °C almost no phage were observed in the supernatant, even in the absence of bacteria, suggesting that they had attached to the filter. A reduction of phage in the supernatant indicated that the phage had attached to the cells, and ended up in the filter rather than the supernatant (Supplementary Fig. 7).

**Toothpick sampling assay**. To assess where in the colonies phage and infected or uninfected PAO1 bacteria were located, the experiments were repeated using 10 replicate PAO1 colonies and 10 mixed (PAO1 and PA14) colonies. We defined four locations to sample from in each colony as shown in Fig. 4a, taking care to sample only from the top of that area (not going so deep as to touch the filter). Note that this is not a very precise method. Errors can result from sampling in

slightly different locations than intended or going too deep with the toothpick and touching the agar, which may contain more phage. Each toothpick was then suspended in 300 μl of PBS, vortexed, and 5 μl of the resulting solution inoculated onto (i) LB agar plates to quantify overall bacterial density, (ii) gentamicin-containing LB agar plates to quantify PAO1 density, (iii) gentamicin-containing LB agar plates saturated with approximately $10^{10}$ PFU/ml of pre-absorbed phages to quantify PAO1 resistance and (iv) TSB + soft agar containing PAO1 as described above to quantify phage density. After 15 h of growth at 37 °C (a fixed duration is critical to make experiments comparable), we imaged each of the resulting spots using a Dino-Lite Edge microscope. A standard curve mapping density to CFUs is shown in Supplementary Fig. 10.

**Microscopy and image analysis**. Images of the colonies were acquired after 12 and 48 h using a Zeiss AXIO Imager M1 fluorescence microscope and a 2.5× objective. PAO1-GFP colonies were imaged using a GFP filter set (excitation: 470/40, emission: 525/50) and PA14-mCherry colonies using an mCherry filter set (excitation: 545/30, emission: 620/60, with automatic exposure), and for mixed colonies, an overlay of the two images was produced using imageJ. Since some colonies after 48 h were too large to fit in one image, a series of 3 × 3 images were acquired and stitched together using autostitch software[65]. Scalebars and white arrows for annotation were added using Inkscape software.

For the toothpick sampling assay, each image was manually cropped to 600 × 600 pixels, converted to grayscale, and a threshold applied using Matlab® R2017b's Image Processing toolbox to yield the photos in Fig. 4b, Supplementary Figs. 8, 9. We then summed the black pixels and white pixels to compute phage or bacterial density, respectively, and divided them by the total number of pixels. The Matlab® code for this image analysis is available upon reasonable request. See supplementary section for additional details.

For the transmission electron microscopy (Fig. 3), the filter and colony were removed with sterile tweezers, placed upside-down and fixed in a 2.5% glutaraldehyde solution (EMS, Hatfield, PA) in phosphate buffer (PB 0.1 M, pH 7.4) for 1 h at room temperature (RT) and post-fixed in a fresh mixture of 1% osmium tetroxide (EMS) with 1.5% of potassium ferrocyanide (Sigma, St. Louis, MO) in PB buffer for 1 h at RT. The samples were then washed twice in distilled water and dehydrated in ethanol solution (Sigma, St Louis, MO, US) at graded concentrations (30% for 40 mins; 50% for 40 mins; 70% for 40 mins; 100% for 2× 30 mins). This was followed by infiltration in 100% Epon resin (EMS, Hatfield, PA, US) overnight, and finally polymerized for 48 h at 60 °C in an oven. Ultrathin sections of 50 nm thick were cut transversally to the filter, using a Leica Ultracut (Leica Mikrosystem GmbH, Vienna, Austria), picked up on a copper slot grid 2× 1 mm (EMS, Hatfield, PA, US) coated with a polystyrene film (Sigma, St Louis, MO, US). Sections were post-stained with uranyl acetate (Sigma, St Louis, MO, US) 4% in water for 10 min, rinsed several times with water followed by Reynolds lead citrate in water (Sigma, St Louis, MO, US) for 10 min and rinsed several times with water. Micrographs were taken with a transmission electron microscope FEI CM100 (FEI, Eindhoven, the Netherlands) at an acceleration voltage of 80 kV with a TVIPS TemCamF416 digital camera (TVIPS GmbH, Gauting, Germany).

**Identifying resistance mutation in PAO1**. PAO1-mutant cells were added to 45 μl of lysis buffer (10 mM TrisHCL, 1 mM EDTA, 0.1% Triton X adjusted to pH 8.0 using NaOH; 2.5 μl of 20 mg/ml solution of lysozyme, Sigma–Aldrich, 62971-10G-F; 2.5 μl of 10 mg/ml proteinase K, Sigma–Aldrich). The sample was lysed using a thermocycler (20 min at 37 °C, 20 min at 55 °C, 20 min at 95 °C). The *galU* gene was amplified from the lysate with forward (5′-CCGACAAGGAAAAGTACCT GG-3′) and reverse (5′-CGCTTGCCCTTGAACTTGTAG-3′) primers. The reaction mixture (25 μl, final volume) contained 15.375 μl of nuclease-free water, 5 μl of 5× Gotaq buffer (Promega M792A), 1 μl of 10 μM forward primer, 1 ul of 10 μM reverse primer, 0.5 μl of 10 μM dNTP mix (Promega U151B), 1U of GoTaq G2 DNA polymerase (Promega, M784B) and 2 μl of bacterial lysate. The PCR was performed with a thermocycler: 5 min of initial denaturation at 95 °C, followed by 35 cycles of denaturation (30 s at 95° C), annealing (30 s at 55 °C), and extension (50 s at 72 °C), with a final extension step (8 min at 72 °C). Amplified products from all samples were verified by gel electrophoresis (Supplementary Fig. 12).

**Statistics and reproducibility**. Each experiment was performed using three technical replicates per condition. Due to this low replicate number, we compared treatments using two-tailed *t*-tests. Experiments were then repeated on separate occasions, and results are reported in supplementary material. We combined data from corresponding treatments across experiments by fitting a linear model to the data and applying a blocked ANOVA test. To test whether phage and bacterial densities correlated in the toothpick assay, we used Pearson's correlation test.

**Reporting summary**. Further information on research design is available in the Nature Research Reporting Summary linked to this article.

## Data availability

All data shown in the manuscript and supplementary information can be found here in ref. [66].

## Code availability

Code for statistical or image analysis, and the mutant PAO1 strain are available from the corresponding author upon reasonable request.

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

## Acknowledgements

We would like to thank Harald Brüssow, Kevin Foster, Flor Arias-Sànchez and Shawna McCallin for insightful feedback and discussions. We thank Kevin Foster for the bacterial strains, Dieter Haas posthumously for the phage, Marc Garcia-Garcerà for extracting the DNA and running a PCR on the mutant strain, and Semhar Ghebrehiwet Tekle for constructing the mCherry plasmid. We appreciate the assistance and support of Damien De Bellis and the Electron Microscopy Facility (EMF) at the University of Lausanne. S.T. and P.P. were funded by the University of Lausanne, F.O. by an SNF Early Postdoc. Mobility grant, S.B. and S.M. by ERC Starting grant 715097.

## Author contributions

S.T., S.B., and P.P. carried out the experiments. F.O., P.P. and S.M. supervised the work. F.O., G.R. and S.M. conceived of the study. S.M. wrote the manuscript. All authors reviewed the manuscript.

## Competing interests

The authors declare no competing interests.
