## [Peer Review File · Communications Biology]

REVIEWERS' COMMENTS:

Reviewer #1 (Remarks to the Author):

The authors investigate how the ecological interactions between closely related bacterial strains can limit the evolution of phage resistance. They demonstrate that this is likely driven by competition reducing the effective population size, and therefore mutation supply of the sensitive strain. They also find that spatial structure also reduces phage resistance by leading to growth arrest at the centre of a bacterial colony. This growth arrest reduces the replication rate of the phage to below that required for a successful epidemic. While many of these ideas have been represented individually, the authors do a good job of combining these factors into a fairly meaningful ecological context. The combination of empirical data, fluorescent imaging and electron microscopy is used well to interrogate the underlying mechanism.

I have one comment:

Lines 142 – 145: “We hypothesized that the presence of PA14 prevented PAO1 from increasing its population size, thereby decreasing its potential to evolve resistance to the phage and survive the treatment.”

The competitor strain reduces the evolution of resistance, but population size isn't the only factor here. By varying population size you test the effect of mutation supply and find that below a certain threshold (10^4 CFU / mL) resistance does not evolve, drawing the conclusion that resistance is solely a function of population size. A non-mutually exclusive alternative hypothesis is that resistance is traded-off against competitive fitness. And this trade-off is only realized once competitor densities are high enough. I am concerned that this is ignored, particularly as line 337 states that the observed resistance mechanism is associated with a trade-off in the form of LPS loss.

My previous comments on the manuscript have been addressed in the rebuttal letter-particularly the statistical analysis.

Reviewer #2 (Remarks to the Author):

The work is interesting and has novel aspects.

Generally, the rebuttal is strong and additional work has been done where necessary.

My only remaining issues are as follows:

1. Fig. 1. The numbering system does not make sense (ie. the order and relative positions of A – F). In addition, red looks like purple to me and I also wonder why no error bars are shown (as did a reviewer)? I could see details of replicates described for the colony experiments but not the liquid ones. If Fig 1 is a plot of all replicates (raw data) as the authors suggest, then it is hard to tell, but it looks like three biological replicates (each plotted individually), but no technical replicates. Is that correct? The authors need to clarify this.

2. Relatedness of PA14 and PAO1. These are both *P.aeruginosa*, but within the wider *P.aeruginosa*, these two strains are not “closely related”. The reference cited Bruggemann et al is probably not the best choice, but it does contain a tree that conforms to the general structure of the majority of *P.aeruginosa* strains being sub-divided into two major clades. A better example would be Freschi et al

2019 GBE 11:109-120 where Figure 2 identifies these two major clades as Group 1 and Group 2. It is even clearer in Freschi et al. 2018 FEMS Micro Letters 365:fny120, where Group 1 and Group2 are clearly identified and the strains are labelled (see Fig. 1), with PAO1 in Group 1 and PA14 (labelled as UBCPP-PA14) in Group 2. This latter reference would be the ideal one, if only one is used, because it includes the PAO1 and PA14 strains on the labelled figure.

Reviewer #1 (Remarks to the Author):

The authors investigate how the ecological interactions between closely related bacterial strains can limit the evolution of phage resistance. They demonstrate that this is likely driven by competition reducing the effective population size, and therefore mutation supply of the sensitive strain. They also find that spatial structure also reduces phage resistance by leading to growth arrest at the centre of a bacterial colony. This growth arrest reduces the replication rate of the phage to below that required for a successful epidemic. While many of these ideas have been represented individually, the authors do a good job of combining these factors into a fairly meaningful ecological context. The combination of empirical data, fluorescent imaging and electron microscopy is used well to interrogate the underlying mechanism.

Thank you!

I have one comment:

Lines 142 – 145: “We hypothesized that the presence of PA14 prevented PAO1 from increasing its population size, thereby decreasing its potential to evolve resistance to the phage and survive the treatment.”

The competitor strain reduces the evolution of resistance, but population size isn't the only factor here. By varying population size you test the effect of mutation supply and find that below a certain threshold (10^4 CFU / mL) resistance does not evolve, drawing the conclusion that resistance is solely a function of population size. A non-mutually exclusive alternative hypothesis is that resistance is traded-off against competitive fitness. And this trade-off is only realized once competitor densities are high enough. I am concerned that this is ignored, particularly as line 337 states that the observed resistance mechanism is associated with a trade-off in the form of LPS loss.

It's true that we had not considered the possibility that phage-resistant mutants would also be less competitiveness against PA14. That said, these mutants would still survive phage attack, but would not be able to grow much due to PA14. We would then not expect them to be eliminated from the population completely, unless PA14 could explicitly kill these cells, which appears unlikely to us in liquid culture. In other words, the only explanation that we can think of for the complete absence of any PAO1 cells in liquid is that they were all killed by phage, i.e. they were sensitive.

My previous comments on the manuscript have been addressed in the rebuttal letter-particularly the statistical analysis.

Reviewer #2 (Remarks to the Author):

The work is interesting and has novel aspects.

Generally, the rebuttal is strong and additional work has been done where necessary.

Thank you!

My only remaining issues are as follows:

1. Fig. 1. The numbering system does not make sense (ie. the order and relative positions of A – F). In addition, red looks like purple to me and I also wonder why no error bars are shown (as did a reviewer)? I could see details of replicates described for the colony experiments but not the liquid ones. If Fig 1 is a plot of all replicates (raw data) as the authors suggest, then it is hard to tell, but it looks like three biological replicates (each plotted individually), but no technical replicates. Is that correct? The authors need to clarify this.

First, thank you for noticing a typo. We changed the color of all the plots from red to purple (to make them more readable for color-blind people), but forgot to change the captions. It now says “purple”.

Second, it's true that the order of the panels is not spatially sequential, but it follows the order in which we refer to them in the text. We believe this is more important. We could alternatively put panels E and F below the first four, but this would make the figure larger overall. We leave it up to the editors to instruct us whether this is necessary.

Finally, the figure shows three technical replicates. We have now clarified this in the figure caption: “All panels show raw data coming from three technical replicates.”

2. Relatedness of PA14 and PAO1. These are both *P.aeruginosa*, but within the wider *P.aeruginosa*, these two strains are not “closely related”. The reference cited Bruggemann et al is probably not the best choice, but it does contain a tree that conforms to the general structure of the majority of *P.aeruginosa* strains being sub-divided into two major clades. A better example would be Freschi et al 2019 GBE 11:109-120 where Figure 2 identifies these two major clades as Group 1 and Group 2. It is even clearer in Freschi et al. 2018 FEMS Micro Letters 365:fny120, where Group 1 and Group2 are clearly identified and the strains are labelled (see Fig. 1), with PAO1 in Group 1 and PA14 (labelled as UBCPP-PA14) in Group 2. This latter reference would be the ideal one, if only one is used, because it includes the PAO1 and PA14 strains on the labelled figure.

Thank you for pointing out these papers. We now cite Freschi et al 2018, and have changed the text to put less emphasis on the relatedness between our two strains: “Since phage are so specific, we believe the choice of a phage-insensitive strain of the same species (Freschi et al. 2018) to be a realistic one.”